# METADD: BOOSTING DATASET DISTILLATION WITH NEURAL NETWORK ARCHITECTURE-INVARIANT GENERALIZATION

## ABSTRACT

Dataset distillation (DD) entails creating a refined, compact distilled dataset from a large-scale dataset to facilitate efficient training. A significant challenge in DD is the dependency between the distilled dataset and the neural network (NN) architecture used. Training a different NN architecture with a distilled dataset distilled using a specific architecture often results in diminished trainning performance for other architectures. This paper introduces MetaDD, designed to enhance the generalizability of DD across various NN architectures. Specifically, MetaDD partitions distilled data into meta features (i.e., the data's common characteristics that remain consistent across different NN architectures) and heterogeneous features (i.e., the data's unique feature to each NN architecture). Then, MetaDD employs an architecture-invariant loss function for multi-architecture feature alignment, which increases meta features and reduces heterogeneous features in distilled data. As a low-memory consumption component, MetaDD can be seamlessly integrated into any DD methodology. Experimental results demonstrate that MetaDD significantly improves performance across various DD methods. On the Distilled Tiny-Imagenet with Sre2L (50 IPC), MetaDD achieves cross-architecture NN accuracy of up to 30.1%, surpassing the second-best method (GLaD) by 1.7%.

## 1 INTRODUCTION

Neural networks (NNs) rely heavily on data, and their performance is directly influenced by the scale and quality of the data Devlin et al. (2018); Ramesh et al. (2022). However, as datasets grow, the cost of training NNs increases significantly. Dataset distillation (DD) Cui et al. (2022) addresses this issue by compressing datasets, producing a smaller, distilled version that can serve as an efficient substitute. This technique is particularly valuable in environments with limited memory and computational resources, such as edge devices or real-time applications. With DD, researchers can achieve competitive model performance while significantly reducing both data size and resource consumption.

A significant challenge in DD is its limited transferability across different NN architectures. Distilled datasets tailored for specific NNs often experience notable performance drop when applied to other architectures. Existing solutions to this cross-architecture gap have various shortcomings. Some do not effectively integrate with vision transformers Zhong & Liu (2023), others only enhance performance for the architectures directly involved in the distillation process Zhou et al. (2024a), and some incur high memory costs due to reliance on image generators Cazenavette et al. (2023). These challenges underscore the need for more robust and versatile DD methods that can maintain strong performance across diverse architectures.

To investigate the causes of DD's cross-architecture gap, we visualize NN architecture biases in feature preferences using Class Activation Maps (CAMs) Zhou et al. (2016); Selvaraju et al. (2017). CAMs highlight the image regions most relevant to a network's predictions, revealing the key features that drive its decision-making. We define the overlap between CAM regions from different architectures as the shared decision-making consensus (meta features), while the non-overlapping regions represent each architecture's unique feature preference (heterogeneous features). Our investigation focuses on two key aspects: (1) why the original dataset shows better cross-architecture

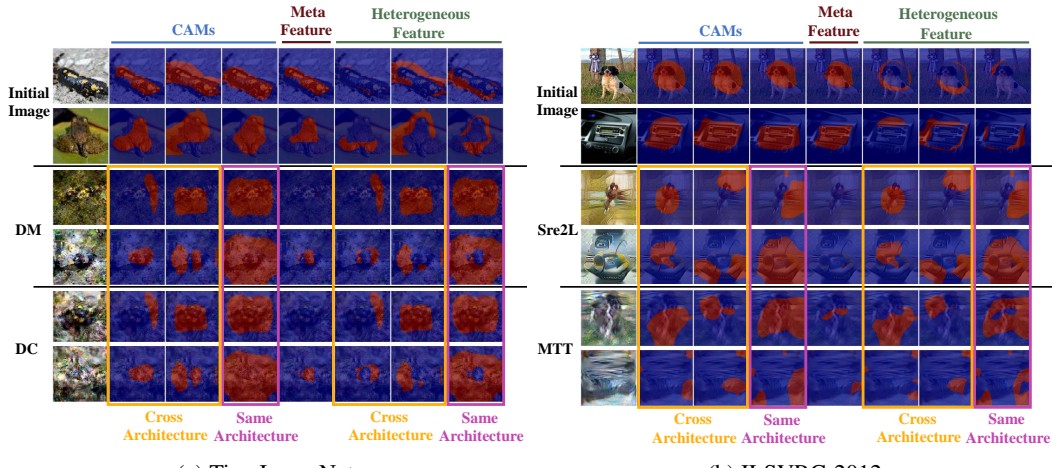

(a) Tiny ImageNet        (b) ILSVRC-2012

Figure 1: Meta and heterogeneous features based on CAM. For Tiny-ImageNet, we utilize DM Zhao & Bilen (2023) and DC Zhao et al. (2020), while MTT Cazenavette et al. (2022) and Sre2L Yin et al. (2024) are employed for ISRL2012. The meta features of an image represent the overlapping areas across different NN CAMs, whereas the heterogeneous features are the unique portions remaining after the meta features are excluded. The synthetic images of every DD method are from the same class as the initial images. ResNet18 is used for distillation, with GoogLeNet and AlexNet serving as cross-structural models.

performance than the distilled dataset, and (2) why the distilled NN architecture outperforms other architectures on the distilled dataset.

By exploring both differences, we find that ***distilled datasets have massive heterogeneous features of the distilled architecture and rare meta features***. As shown in Figure 1: (1) Original images show substantial meta features across different architectures trained on the original dataset, whereas distilled images show almost no meta features. This meta feature comparison explains why original dataset performs more consistently across different NNs than distilled data. (2) Distilled images have significantly more heterogeneous features with the same architecture used for DD compared to crossed architectures. These heterogeneous features represent the same architecture can extract more semantic information from distilled data, which crossed NN architecture cannot capture, leading to cross-architecture gap.

Based on the above observations, we propose MetaDD to improve the cross-architecture generalization of DD. MetaDD begins with using an architecture-invariant loss to obtain and maximize the exposure of diverse features across different NN architectures. Then, MetaDD decouples heterogeneous and meta features by transferring distilled data's CAMs to a common space. By driving the evolution of distilled data towards maximizing meta features, MetaDD encourages to form a generalized consensus cross different NN architectures. MetaDD maintains low memory consumption by persistently freezing various NN architectures during DD. We conducted comprehensive experiments using DC, DM, MTT, and Sre2L as baselines. Incorporating MetaDD improved DD performance in cross-architecture training. MetaDD covers typical NN architectures, ensuring that even unconsidered models benefit from the meta features generated. On TinyImagenet and ILSVRC-2012, MetaDD has an average accuracy increase of 1.6% and 1.0% compared with GlaD Cazenavette et al. (2023).

## 2 RELATED WORK

### 2.1 DATASET DISTILLATION

DD Liu et al. (2023); Sajedi et al. (2023); Du et al. (2024) generates distilled datasets by aligning distilled data with the original data using a specific NN. This alignment is achieved through various techniques. For instance, model gradients Zhao et al. (2020) are used to adjust the distilled data to match the gradient patterns of the original data. Similarly, the features Zhao & Bilen (2023)

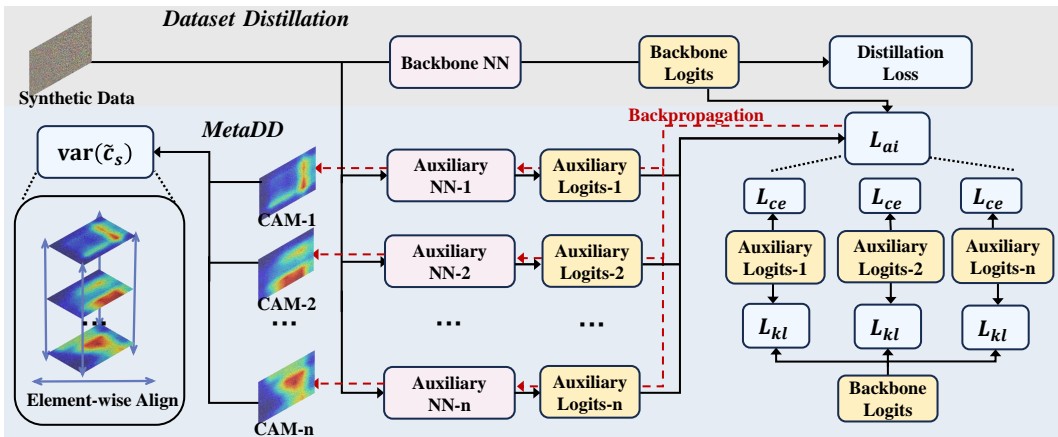

Figure 2: The framework of MetaDD. Our method is designed to supervise the synthesis of data during training to ensure it exhibits low-variance CAMs across multiple pre-trained NNs.

extracted by the NN are aligned to ensure that the distilled data captures the same feature distributions as the original data. Moreover, parameter trajectories Cazenavette et al. (2022) are tracked and matched, providing a dynamic way to align the evolving parameters during training. Additionally, kernel ridge regression statistics of the NNs Nguyen et al. (2020; 2021) are aligned to refine the distilled data further, ensuring that it statistically mirrors the original data from the kernel-based perspective.

Furthermore, the creation of novel data augmentation components Zhao & Bilen (2021); Liu et al. (2022a); Zhou et al. (2024c) significantly enhances the distilled datasets. These components introduce new data transformations that enrich the variability of the distilled data. Reusable distillation paradigms He et al. (2024); Yang et al. (2023); Shang et al. (2024) are also employed, which involve extracting and transferring knowledge from multiple models or datasets into the distilled dataset, thus improving its performance. Additionally, patches for fixing defects Lee et al. (2022); Cui et al. (2023); Du et al. (2023); Zhao et al. (2023) are developed and applied, addressing any inconsistencies or errors in the distilled data. These patches help maintain the distilled datasets' integrity and accuracy.

## 2.2 CLASS ACTIVATION MAPPING

CAM Kundu (2020); Wang et al. (2020a); Selvaraju et al. (2017); Fu et al. (2020); Jiang et al. (2021); Wang et al. (2020b); Chattopadhay et al. (2018); Omeiza et al. (2019) algorithms constitute an important class of methods in the field of deep learning, aimed at enhancing the interpretability of NNs. The original CAM Zhou et al. (2016) involves modifying the network architecture to connect directly from the global average pooling layer to the output layer, allowing the model to highlight important areas of the image for predictions of specific classes. Subsequent developments, such as Grad-CAM Selvaraju et al. (2017), offer a universal solution that does not require modifications to the network architecture. Additionally, variants such as Grad-CAM++ Chattopadhay et al. (2018) and Score-CAM Wang et al. (2020b) have further improved the accuracy and robustness of the heatmaps.

Although CAM was initially designed for CNNs, researchers have begun exploring how similar concepts can be applied to transformer-based visual models. To achieve this application, CAMs are generated by analyzing the attention weights from the last layer Sun et al. (2023). Consequently, researchers have begun developing new techniques and approaches to improve the vision transformer CAMZhu et al. (2023); Xu et al. (2022), for example, by adjusting or combining attention weights from different layers, or developing specialized interpretative modules to produce clearer and more meaningful visual explanations. facilitating further optimization of NN architectures and interpretability improvements.

## 3 METHOD

In this section, we detail the generalization component proposed to mitigate cross-structural performance losses in DD. An overview of our method is illustrated in Figure 2.

### 3.1 PRELIMINARIES

Suppose there is an original dataset $T = \{(x_1, y_1), \ldots, (x_{|T|}, y_{|T|})\}$ with $|T|$ pairs of training samples $x_t$ and corresponding labels $y_t$. The goal of DD is to synthesize a dataset $S = \{(\hat{x}_1, \hat{y}_1), \ldots, (\hat{x}_{|S|}, \hat{y}_{|S|})\}$ where $|S| \ll |T|$. The model trained on $S$ is expected to perform similarly to one trained on $T$. For a set of different NN architectures $\theta = \{\vartheta_1, \ldots, \vartheta_{|M|}\}$, we define the performance loss when a distilled dataset $S(\vartheta_u)$ distilled on $\vartheta_u$ is used to train $\vartheta_v$ as $\Delta \text{Acc}(\vartheta_v|\vartheta_u) = \text{Acc}_S(\vartheta_v)^{\vartheta_v} - \text{Acc}_S(\vartheta_u)^{\vartheta_v}$. $\text{Acc}_S(\vartheta_v)^{\vartheta_v}$ denotes the accuracy obtained by distilling and training on the same model architecture $\vartheta_v$. Our objective is to minimize the total cross-structural performance loss for all $\vartheta_u, \vartheta_v \in \theta$: $\min \sum_{i=1}^{|M|} \sum_{j \neq i}^{|M|} \Delta \text{Acc}(\vartheta_v|\vartheta_u)$.

### 3.2 HETEROGENEOUS AND META FEATURES

For data's heterogeneous and meta features' visual presence, we initially use Grad-CAM to capture the CAM $C_s = \bigcup_m^{|M|} \{c_s^m\}$ of the data $\hat{x}_s$ across different pre-trained model architectures $\theta$. We then interpolate the CAMs of the data to the same size, and each matrix element of all CAMs is normalized to a range from 0 to 1. The processed CAMs is $\tilde{C}_s = \bigcup_m^{|M|} \{\tilde{c}_s^m\}$. To mitigate the influence of random factors, we disregard the low-confidence activation regions within the CAMs, specifically those areas where the values are below 0.5. For high-confidence CAMs, we define the pixel locations of the meta features as the overlapping sections across all NNs' CAMs. The mask representing the meta feature is denoted as :

$$\mu_s = \prod_m^{|M|} H(\tilde{c}_s^m), H(c)_{i,j} = \begin{cases} 1 & \text{if } c_{i,j} \geq 0.5 \\ 0 & \text{otherwise} \end{cases} \tag{1}$$

Subtracting the common feature areas from each CAM yields each architecture's heterogeneous areas of focus. The mask of heterogeneous feature areas of the NN architecture $\vartheta_u$ in the distilled image $x_s$ is represented as:

$$\beta_s^m = H(\tilde{c}_s^m) - \mu_s \tag{2}$$

Further experiments using TinyImagenet confirm that our defined heterogeneous and meta features respectively exhibit specific and common preferences across different NN architectures. We initially erase the heterogeneous feature pixels corresponding to different pre-trained NN architectures in TinyImagenet. Architectures are ResNet34, MobileNetV2, GoogleNet, VGG19, EfficientNet, and ViT. Then we train ViT from scratch with the erased TinyImagenet. The accuracy differenece in Figure 3 indicates that ViT suffers the most when losing self-heterogeneous features and the least loss when losing other architectures' heterogeneous

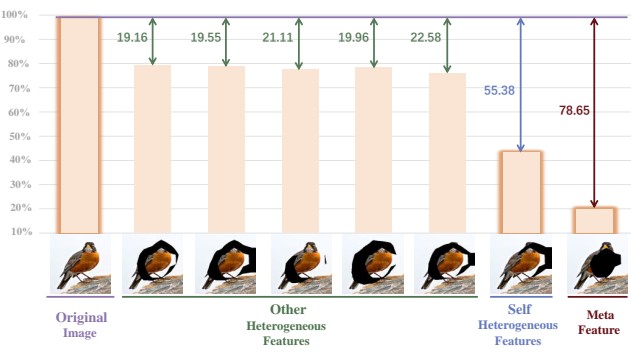

Figure 3: ViT's validation accuracy on different erased TinyImagenet. The numbers represent the difference in accuracy between the erased and the original dataset.

features. We perform the same experiment by erasing the meta features of all architectures in TinyImagenet. Result shows ViT experience significant perfor-

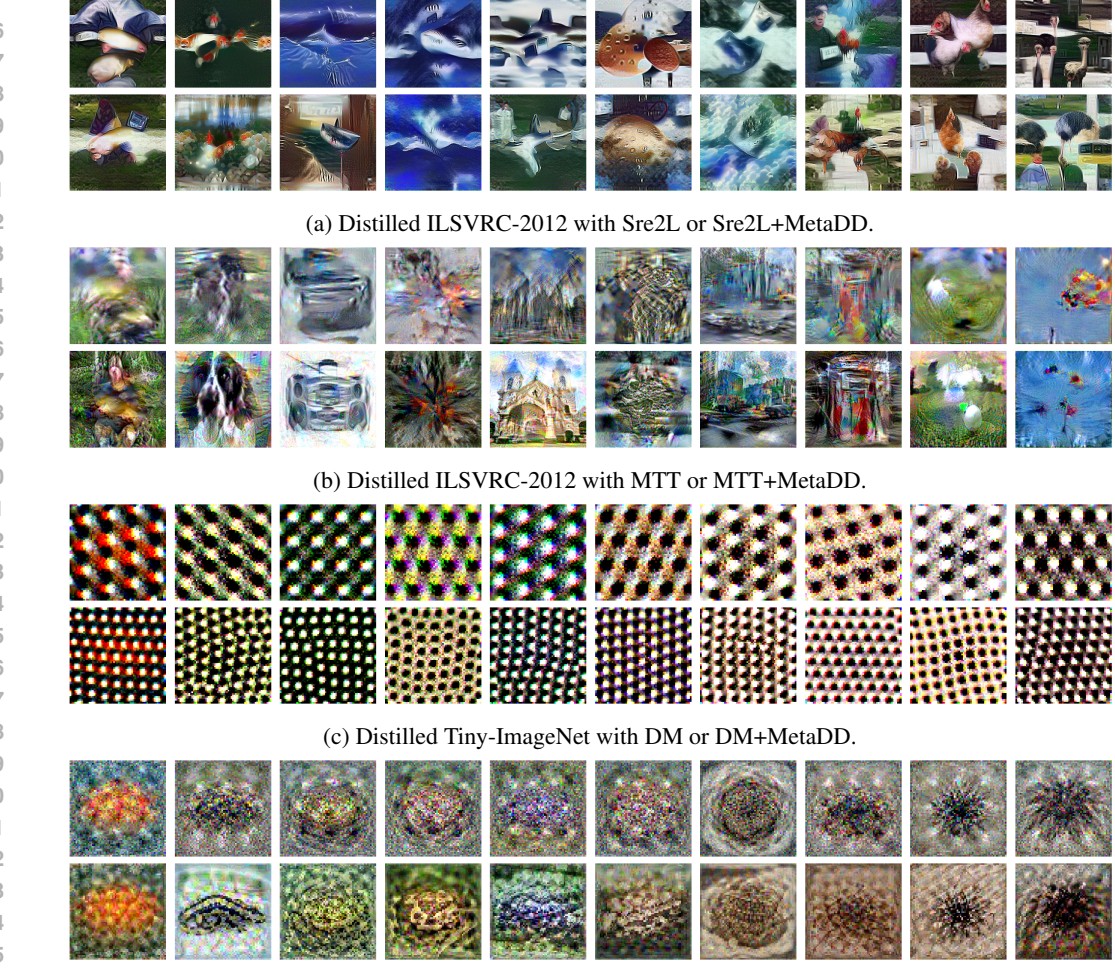

(a) Distilled ILSVRC-2012 with Sre2L or Sre2L+MetaDD.

(b) Distilled ILSVRC-2012 with MTT or MTT+MetaDD.

(c) Distilled Tiny-ImageNet with DM or DM+MetaDD.

(d) Distilled Tiny-ImageNet with DC or DC+MetaDD.

Figure 4: In each subplot, the first row displays images generated by the original DD algorithm, while the second row presents images generated after integrating MetaDD.

mance drops after losing meta features. These experiments thoroughly illustrate the distinct natures of heterogeneous and meta features and lay the empirical foundation for our method.

### 3.3 METADD

MetaDD define the NN used in the original DD method as the backbone NN $\vartheta_\mu$. To obtain a distilled dataset that generalizes across architectures, MetaDD incorporates pre-trained NNs of various other architectures, referred to as auxiliary networks:

$$\theta = \{\vartheta_1, \ldots, \vartheta_m, \ldots, \vartheta_{|M|}\} \tag{3}$$

MetaDD employs an architecture-invariant loss function to backpropagate and obtain CAMs on distilled data for different auxiliary networks and then normalizes these CAMs. By reducing the variance at the same locations across these CAMs, the CAMs tend to be similar. Through multiple rounds of updates, MetaDD enhances the test performance of the distilled dataset on other NNs.

**Architecture-invariant Loss**. We use a mixed loss function of cross-entropy and KL divergence to obtain the Grad-CAM images, specifically expressed as:

$$L_{ai} = \sum_m^{|M|} L_{ce}\left(\vartheta_m\left(x\right), y\right) + \sum_m^{|M|} L_{kd}\left(\vartheta_m\left(x\right), \vartheta_\mu\left(x\right)\right) \tag{4}$$

$$L_{kd}\left(\vartheta_m\left(x\right), \vartheta_\mu\left(x\right)\right) = \vartheta_m\left(x\right) \log \frac{\vartheta_m\left(x\right)}{\vartheta_\mu\left(x\right)} \tag{5}$$

---

**Algorithm 1** MetaDD Algorithm

---

**Require:** Training set $T$, Randomly initialized set of distilled samples $S$, backbone NN $\vartheta_\mu$, auxiliary NNs $\theta$, training iterations $K$, learning rate $\eta$

1: **for** $k = 0, \ldots, K - 1$ **do**
2:    Sample mini-batch pairs $B_s \in S$ and $B_t \in T$
3:    Compute $L_{all} = L_{dd}(\vartheta_\mu, B_s, B_t)$
4:    Compute $L_{ai}(\vartheta_m, B_s, \theta)$ from Equation 4.
5:    $L_{all} = L_{all} + L_{ai}(\vartheta_m, B_s, \theta)$
6:    **for all** $(\hat{x}_s, \hat{y}_s) \in B_s$ **do**
7:       $\tilde{c}_s \leftarrow \emptyset$
8:       **for all** $\vartheta_m \in \theta$ **do**
9:          Compute the cam $\tilde{c}_s^m$ from Equation 6 or 8
10:         $\tilde{c}_s := \tilde{c}_s^m \cup \tilde{c}_s$
11:       Compute $L_{pos}$ from Equation 9
12:       Compute var $(\tilde{c}_s)$ from Equation 11
13:       $L_{all} = L_{all} + \text{var}(\tilde{c}_s) + L_{pos}$
14:    Update $B_s \leftarrow B_s - \eta \frac{\partial L_{all}}{\partial B_s}$
     **return** $S$

---

Compared to solely utilizing cross-entropy loss for backpropagation to obtain CAMs from auxiliary NNs and aligning these CAMs, the architecture-invariant Loss, which includes an additional KL divergence loss, offers significant advantages: the CAMs generated by architecture-invariant loss reflect the distilled data features that need to be focused on when transferring knowledge from the auxiliary NNs to the main NN. Consequently, architecture-invariant loss maximumly displays heterogeneous features antagonistic to the main NN, which will be transferred to meta features.

**Modified Class Active Map**. We utilize a modified Grad-CAM Selvaraju et al. (2017) to obtain activation maps from various convolutional NNs. We initially perform a forward pass to acquire the unflattened feature maps $A$ from the last fully connected layer. Let $A^k$ represent the feature map activations of the $k$-th channel for $A$. Then MetaDD computes the gradient of $L_{ai}$, concerning feature map activations $A^k$. These gradients flowing back are global-average-pooled to obtain the neuron importance weights $\alpha_k^c$. Then, the linear combination of these weighted activation maps gives the class-discriminative localization map $c$ used to highlight the important regions.

$$\alpha_k = \frac{1}{I * J} \sum_i^I \sum_j^J \frac{\partial L_{ai}}{\partial A_{ij}^k}, \quad c = \left( \sum_k \alpha_k A^k \right) \tag{6}$$

where $i$ and $j$ are the spatial dimensions of the feature map, and $Z$ is the total number of elements in the feature map.

For Vision Transformers, we consider the output of the last transformer layer $\mathbf{A} \in \mathbb{R}^{N \times D}$, where $N$ is the number of patches and $D$ is the dimension of features per patch. The class token's output $\mathbf{Z}$ is utilized by an MLP head to generate class predictions $y^c$. Attention scores are computed as:

$$W_{\text{agg}} = \sum_{\text{heads}} W_{\text{head}}[\text{cls}, :] \tag{7}$$

where $\mathbf{Q}$ and $\mathbf{K}$ are the query and key matrices from the multi-head self-attention mechanism. The CAM is generated by:

$$W_{\text{agg}} = \sum_{\text{heads}} W_{\text{head}}[\text{cls}, :], \quad c = (W_{\text{agg}} \cdot \mathbf{Z}) \tag{8}$$

where $W_{\text{agg}}$ is the aggregated attention across heads. More different from Grad-CAM, we do not employ the ReLU function in Equation 6 and 8. This is because the negative parts of the activation maps are also essential for our optimization. Meanwhile, we ensure that the positive values in all different CAMs are maximized as much as possible:

$$L_{pos} = \sum_i \sum_j c_{i,j} \tag{9}$$

Table 1: The cross-architecture generalization experiments on ILSVRC-2012 and Tiny-ImageNet. $L_{ai}$ is DD using architecture-invariant loss function without generating CAMs.

| | | ILSVRC-2012 (IPC = 10) | | | | | | | | |
|---|---|---|---|---|---|---|---|---|---|---|
| | | Auxiliary/Seen | | | | Unseen | | | | |
| Method | Component | ResNet34 | MobileNetV2 | GoogleNet | ViT-B-16 | AlexNet | ResNet50 | Vgg19 | Swin-S | Average |
| TesLa | none | 11.8±1.3 | 9.6±1.1 | 10.8±0.6 | 11.2±1.7 | 9.2±1.2 | 11.7±0.6 | 10.8±0.9 | 10.3±0.7 | 10.6 |
| | Dream Liu et al. (2023) | 12.1±1.1 | 9.9±1.2 | 10.9±0.3 | 11.4±1.2 | 9.6±0.7 | 11.4±0.3 | 10.3±0.7 | 10.6±0.4 | 10.8 |
| | GLaD Cazenavette et al. (2023) | 12.8±1.1 | 11.1±0.6 | 11.9±1.1 | 12.1±0.3 | 11.7±1.2 | 12.4±1.3 | 12.9±0.6 | 11.7±1.1 | 12.1 |
| | MetaDD | **13.1**±0.3 | **13.4**±0.2 | **14.2**±0.3 | **12.9**±0.6 | **12.4**±0.2 | **13.2**±0.1 | **13.7**±0.2 | **11.9**±0.5 | 13.1 |
| Sre2L | none | 13.3±0.1 | 12.1±0.3 | 12.7±0.3 | 13.7±0.3 | 12.9±0.8 | 11.8±0.2 | 11.9±0.2 | 13.1±0.6 | 12.9 |
| | Dream Liu et al. (2023) | 13.5±0.7 | 12.3±0.5 | 12.9±0.3 | 13.7±0.5 | 13.2±0.7 | 12.2±0.3 | 12.2±0.3 | 13.3±0.5 | 13.2 |
| | GLaD Cazenavette et al. (2023) | 14.6±0.2 | 13.8±0.2 | 14.2±0.2 | 14.6±1.2 | 13.6±0.2 | 12.9±1.2 | 13.9±1.2 | 14.2±0.3 | 13.9 |
| | MetaDD | **14.8**±0.3 | **14.9**±0.1 | 13.8±0.6 | 14.2±0.4 | **14.8**±0.4 | **13.9**±0.2 | **15.8**±0.7 | **15.9**±0.1 | 14.6 |

(a) ILSVRC-2012

| | | Tiny-ImageNet (IPC = 50) | | | | | | | | |
|---|---|---|---|---|---|---|---|---|---|---|
| | | Auxiliary/Seen | | | | Unseen | | | | |
| Method | Component | ResNet34 | MobileNetV2 | GoogleNet | ViT-B-16 | AlexNet | ResNet50 | Vgg19 | Swin-S | Average |
| DC | none | 11.2±0.4 | 10.8±0.1 | 9.9±0.4 | 11.7±0.4 | 11.0±0.1 | 10.8±0.1 | 10.2±0.1 | 11.4±0.7 | 10.9 |
| | DreamLiu et al. (2023) | 11.5±0.7 | 11.0±0.5 | 10.2±0.4 | 12.1±0.6 | 11.4±0.2 | 11.3±0.1 | 10.5±0.7 | 11.7±0.5 | 11.1 |
| | GLaD Cazenavette et al. (2023) | 13.4±0.3 | 13.5±0.1 | 12.8±0.1 | 12.1±0.2 | 12.3±0.1 | 12.2±0.1 | 11.0±0.2 | 12.4±0.1 | 12.5 |
| | MetaDD | **13.6**±0.1 | **14.1**±0.1 | **14.7**±0.2 | **14.7**±0.1 | **13.8**±0.4 | **14.9**±0.3 | **12.6**±0.1 | **13.7**±0.6 | 13.8 |
| DM | none | 10.9±0.2 | 11.4±0.1 | 10.6±0.7 | 11.3±0.2 | 11.9±0.3 | 10.1±0.4 | 10.7±0.4 | 11.8±0.4 | 11.2 |
| | Dream Liu et al. (2023) | 11.2±0.2 | 11.6±0.4 | 10.9±0.3 | 11.5±0.4 | 12.1±0.5 | 10.4±0.3 | 10.9±0.5 | 12.0±0.6 | 11.4 |
| | GLaD Cazenavette et al. (2023) | 11.6±0.1 | 11.9±0.4 | 11.2±0.1 | 11.8±0.2 | 12.1±0.2 | 11.1±0.3 | 11.8±0.5 | 12.6±0.2 | 11.9 |
| | MetaDD | **12.1**±0.1 | **12.4**±0.3 | **12.5**±0.2 | **14.3**±0.7 | **13.2**±0.2 | **14.1**±0.5 | **14.2**±0.2 | **15.1**±0.4 | 13.5 |

(b) Tiny-ImageNet

**CAM Variance Loss**. After obtaining the heterogeneous CAMs of all auxiliary networks relative to the backbone network, we interpolate and normalize these CAMs to the same size and range:

$$\tilde{c}_s^m = \frac{P\left(c_s^m\right) - \min P\left(c_s^m\right)}{\max P\left(c_s^m\right) - \min P\left(c_s^m\right)} \tag{10}$$

$P\left(\right)$ is the interpolating operation. We then calculate the variance at the same positions in the processed heterogeneous CAMs:

$$\mathrm{var}\left(\tilde{c}_s\right) = \frac{1}{I * J} \sum\nolimits_i^I \sum\nolimits_j^J \left(\frac{1}{M} \sum\nolimits_m^{|M|} \tilde{c}_{s,i,j}^m - \tilde{c}_{s,i,j}^m\right)^2 \tag{11}$$

By minimizing the variance across all positions, the heterogeneous CAMs will tend to be similar. In the process of becoming similar, the features of the data distilled by the backbone NN will be more acceptable to other network architectures. The final loss function of MetaDD is:

$$L_{all}(x_s) = L_{dd} + L_{ai} + \mathrm{var}\left(\tilde{c}_s\right) + L_{pos} \tag{12}$$

$L_{dd}$ is DD method loss function. Through MetaDD, we ensure that the distilled data features are as universal as possible rather than heterogeneous. The process of MetaDD is shown in Algorithm 1.

During the process of obtaining heterogeneous CAMs, the parameters of all different pre-trained NN architectures are frozen. Using pre-trained models with frozen parameters implies low VRAM consumption. Thus, while encompassing multiple different pre-trained NNs, MetaDD still saves computational resources. As a low computational consumption component, MetaDD can be combined with various DD methods to achieve optimal cross-structural training generalization.

## 4 EXPERIMENTS

### 4.1 EXPERIMENTAL SETUP

We evaluated our method (MetaDD) for DD from CIFAR-10 at a resolution of 32 x 32, Tiny-ImageNet Le & Yang (2015) at 64 x 64, and ILSVRC-2012 Deng et al. (2009) at 224 x 224. Our experimental code is based on open-source repositories for DC, DM, Tesla, MTT, and Sre2L. Tesla represents a memory-optimized version of MTT. For each method, we directly integrated MetaDD into the existing codebases. While keeping the hyperparameters of the existing methods unchanged. We show part of distilled images in Figure 4.

Table 2: CIFAR10 cross-architecture average accuracy.

| Method | Component | CIFAR10 | | | |
| | | Auxiliary/Seen (Average) | | Unseen (Average) | |
| | | IPC = 1 | IPC = 10 | IPC = 1 | IPC = 10 |
|---|---|---|---|---|---|
| DC | none | 17.6±1.1 | 38.1±0.3 | 16.1±1.2 | 39.7±1.1 |
| | DreamLiu et al. (2023) | 17.8±1.0 | 38.5±0.7 | 16.5±1.1 | 39.9±1.0 |
| | GLaD Cazenavette et al. (2023) | 21.2±0.4 | 39.1±1.2 | 20.9±1.2 | 39.8±1.2 |
| | MetaDD | **22.1**±1.1 | **42.2**±1.1 | **21.3**±1.2 | **40.3**±0.6 |
| DM | none | 18.9±1.2 | 40.1±1.4 | 17.8±0.8 | 39.8±0.6 |
| | Dream Liu et al. (2023) | 18.9±0.9 | 40.6±1.1 | 17.9±0.7 | 40.3±0.4 |
| | GLaD Cazenavette et al. (2023) | 19.2±0.3 | 41.2±0.5 | 18.9±1.2 | 40.1±1.2 |
| | MetaDD | **20.1**±1.2 | **42.3**±0.7 | **19.2**±0.7 | **40.3**±1.2 |
| MTT | none | 37.2±1.2 | 52.1±2.1 | 36.2±0.4 | 50.9±0.2 |
| | Dream Liu et al. (2023) | 37.6±1.0 | 52.3±2.2 | 36.7±0.1 | 51.2±0.4 |
| | GLaD Cazenavette et al. (2023) | **38.7**±0.2 | 52.2±1.2 | **37.8**±0.3 | 51.4±1.2 |
| | MetaDD | 37.9±1.2 | **53.1**±1.3 | 37.2±0.2 | **52.2**±0.6 |
| Sre2L | none | 41.2±1.4 | 59.8±0.3 | 40.1±1.2 | 58.8±0.5 |
| | Dream Liu et al. (2023) | 41.5±1.2 | 60.2±0.7 | 40.4±1.0 | 59.1±0.3 |
| | GLaD Cazenavette et al. (2023) | 42.1±1.2 | 60.2±1.1 | 42.8±1.7 | 59.7±1.6 |
| | MetaDD | **42.5**±1.0 | **60.4**±0.9 | **44.3**±0.8 | **61.2**±1.2 |

**Baselines**. In addition to MetaDD, we also report the performance of the existing component GLaD Cazenavette et al. (2023) and Dream Liu et al. (2023). GLaD stores distilled data as feature vectors and uses a generator to create high-definition images as inputs during NN training. Dream selects representative original images for DD.

**Neural Architecture**. We employed the ConvNet Gidaris & Komodakis (2018) architecture as our backbone NN for DC, DM, and MTT/Tesla. The Depth-n ConvNet consists of n blocks followed by a fully connected layer. Each block comprises a 3x3 convolutional layer with 128 filters, instance normalization[58], ReLU nonlinearity, and a 2x2 average pooling with a stride of 2. For Sre2L, we use ResNet18 as our backbone NN. we use ResNet34 He et al. (2016), MobileNetV2 Sandler et al. (2018), GoogleNet Szegedy et al. (2015), and ViT-B-16 Dosovitskiy et al. (2020) as our auxiliary NN architectures. All auxiliary NNs are pre-trained using original datasets. We trained the distilled dataset using 8 different NN architectures to test the algorithm's cross-architecture generalizability. In addition to the auxiliary NN architectures, the test also included four architectures not involved in DD: AlexNet Krizhevsky et al. (2017), ResNet50, Vgg19 Simonyan & Zisserman (2014), and Swin-S Liu et al. (2022b).

**Evaluation Metrics**. We evaluated the cross-architecture generalizability of the algorithm by averaging the top-1 accuracy of NNs trained on the distilled dataset on the validation set. This average accuracy measure is a robust indicator of the algorithm's cross-architecture generalizability.

**Training Paradigm**. The training paradigm for all NNs and datasets is consistent: it includes Stochastic Gradient Descent (SGD) with 0.9 momentum, $1e-4$ weight decay, followed by 500 rounds of linear warm-up and then 500 rounds of cosine decay. Each architecture employs an appropriate (fixed) initial learning rate. The training process is repeated three times, and the average validation accuracy $\pm$ one standard deviation is reported.

## 4.2 CROSS-ARCHITECTURE GENERALIZATION

We initially validated our algorithm's capacity for enhancing cross-architecture generalization at ILSVRC-2012 and Tiny-ImageNet. In Table 1, we employed GLaD, ModelPool, and MetaDD to assist DD methods. The results from Table 1 demonstrate that MetaDD effectively reduces overfitting in the backbone NNs. Compared to other baselines, our method generally outperforms in most cases. Moreover, NN architectures included in the auxiliary NN set show similar performance to the unseen NN architectures. Hence, by incorporating specified NN architectures to MetaDD, MetaDD can offer customized services tailored to situations with a specific focus on different NN architectures. Following this, in Table 2, we distilled datasets of varying scales under CIFAR10 and conducted analogous experiments. The results in Table 2 indicate that our method still helps mitigate overfitting. Compared to distillation at higher resolutions, GLaD exhibits weaker performance.

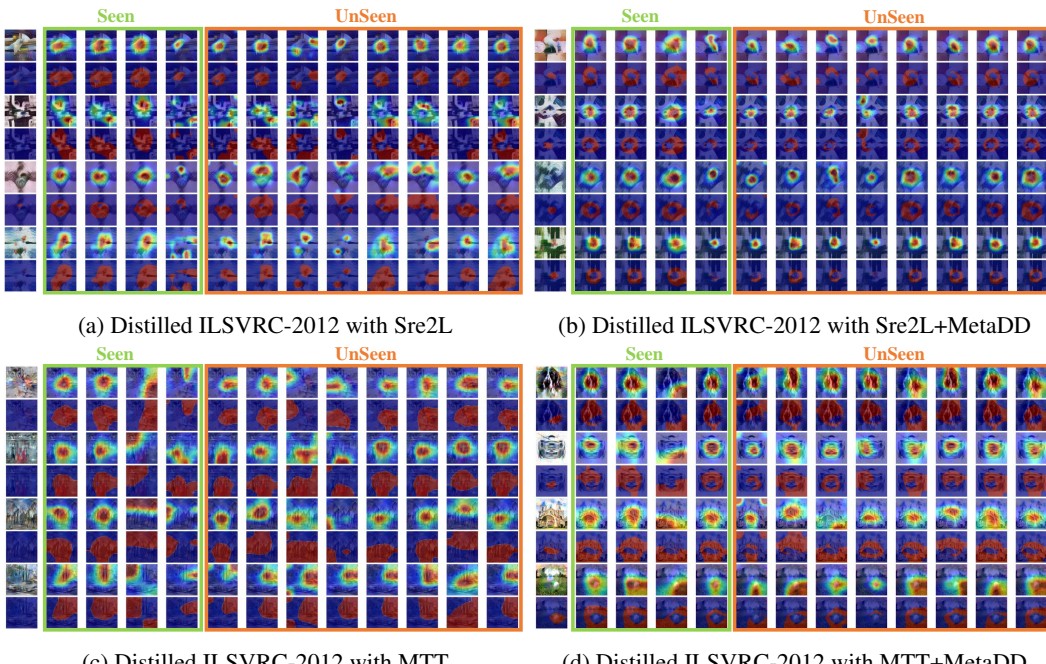

(a) Distilled ILSVRC-2012 with Sre2L

(b) Distilled ILSVRC-2012 with Sre2L+MetaDD

(c) Distilled ILSVRC-2012 with MTT

(d) Distilled ILSVRC-2012 with MTT+MetaDD

Figure 5: Meta Features' Contagious Generalizability visualization.

## 4.3 META FEATURES' CONTAGIOUS GENERALIZABILITY

In the previous section, we demonstrated that MetaDD can enhance performance on NN architectures not included in the auxiliary NNs (unseen NNs). This phenomenon, which we term "contagious generalizability", is attributed to the universality of MetaDD. Figure 5 illustrates the meta features of distilled data across both unseen and seen architectures. In every subfigure, the first image in rows 1, 3, 5, and 7 shows distilled data, followed by CAMs from various architectures trained on the dataset including the distilled data. In rows 1, 3, 5, and 7, CAMs two to five ('seen' frame) correspond to auxiliary architectures used in MetaDD, while CAMs six to thirteen ('unseen' frame) are from architectures not used. The first image in rows 2, 4, 6, and 8 presents meta features from all architectures, with the following images showing heterogeneous features from architectures trained on the distilled data above.

Images generated by the original DD method exhibit almost no meta features. However, with the integration of MetaDD, the distilled images possess meta features recognizable by seen NNs and unseen NNs. This indicates that the meta features introduced by MetaDD are typical and widely applicable, further substantiating the efficacy of MetaDD in enhancing cross-architecture generalizability.

## 4.4 TRAINING COST ANALYSIS

We compare the GPU memory consumption of our method with that of GLaD. We kept all other conditions identical between the two methods. As shown in Table 4, on CIFAR-10, our method reduces memory usage by $\tilde{2}x$ compared to GLaD. The runtime is reduced by $\tilde{0}.3x$. GLaD consumes a significant amount of memory due to the use of generators, whereas our method maintains low memory usage even while accommodating 4 auxiliary NNs. Because the parameters of the auxiliary NNs are always kept frozen, we can scale to a larger number of auxiliary NNs.

| Method | Loss | Accuracy |
|--------|------|----------|
| MTT | None | $52.1 \pm 2.1$ |
| | with $L_{ai}$ | $52.4 \pm 1.3$ |
| | with $L_{pos}$ | $52.3 \pm 0.7$ |
| | with var $(\tilde{c}_s)$ | $52.9 \pm 0.5$ |
| Sre2L | None | $59.8 \pm 0.3$ |
| | with $L_{ai}$ | $59.9 \pm 1.1$ |
| | with $L_{pos}$ | $60.2 \pm 0.2$ |
| | with var $(\tilde{c}_s)$ | $60.4 \pm 0.2$ |

Table 3: Ablation Study on CIFAR10 with IPC=10.

| Method(Dataset) | Component | Memory(GB) | Time (Minutes) |
|---|---|---|---|
| MTT(CIFAR10) | - | 19.9 | 98 |
| | MetaDD | **22.6** | **107** |
| | GlaDCazenavette et al. (2023) | 39.1 | 152 |
| | ModelPoolZhou et al. (2024b) | 32.4 | 219 |
| TesLa(ILSVRC-2012) | - | 68.7 | 1538 |
| | MetaDD | **76.4** | **1601** |
| | GlaDCazenavette et al. (2023) | 119.1 | 1912 |
| | ModelPoolZhou et al. (2024b) | 89.4 | 2125 |

Table 4: Memory cost and training time for different methods on CIFAR10 and ILSVRC-2012 datasets with ipc=10.

| | | ILSVRC-2012 (IPC = 10) | | | | | | | | |
|---|---|---|---|---|---|---|---|---|---|---|
| | | Auxiliary/Seen | | | | Unseen | | | | |
| Quantity | Component | ResNet34 | MobileNetV2 | GoogleNet | ViT-B-16 | AlexNet | ResNet50 | Vgg19 | Swin-S | Average |
| TesLa | none | 11.8±1.3 | 9.6±1.1 | 10.8±0.6 | 11.2±1.7 | 9.2±1.2 | 11.7±0.6 | 10.8±0.9 | 10.3±0.7 | 10.7 |
| | 1 | 12.1±1.1 | 10.2±0.6 | 11.3±1.1 | 11.7±0.3 | 11.7±1.2 | 12.1±1.3 | 11.4±0.6 | 10.8±1.1 | 11.5 |
| | 2 | 12.4±1.1 | 11.4±0.6 | 12.1±1.1 | 12.1±0.3 | 11.9±1.2 | 12.5±1.3 | 11.9±0.6 | 11.2±1.1 | 12.3 |
| | 3 | 12.7±0.2 | 12.1±0.3 | 13.8±0.4 | 13.1±0.5 | 12.0±0.1 | 12.9±0.1 | 12.7±0.2 | 11.5±0.5 | 12.8 |
| | 4 | **13.1**±0.3 | **13.4**±0.2 | **14.2**±0.3 | **12.9**±0.6 | **12.4**±0.2 | **13.2**±0.1 | **13.7**±0.2 | **11.9**±0.5 | 13.4 |
| Sre2L | none | 13.3±0.1 | 12.1±0.3 | 12.7±0.3 | 12.3±0.3 | 12.9±0.8 | 11.8±0.2 | 11.9±0.2 | 13.1±0.6 | 12.5 |
| | 1 | 13.9±0.2 | 12.7±0.2 | 12.7±0.2 | 12.9±1.2 | 12.7±0.2 | 13.2±1.2 | 12.9±1.2 | 13.9±0.3 | 13.1 |
| | 2 | 14.1±0.2 | 13.3±0.2 | 13.0±0.2 | 13.2±1.2 | 13.3±0.2 | 13.6±1.2 | 13.4±1.2 | 14.2±0.3 | 13.8 |
| | 3 | 14.5±0.3 | 14.3±0.3 | 13.4±0.4 | 13.5±0.2 | 14.0±0.3 | 13.9±0.1 | 14.3±0.4 | 15.1±0.3 | 14.6 |
| | 4 | **14.8**±0.3 | **14.9**±0.1 | **13.8**±0.6 | **14.2**±0.4 | **14.8**±0.4 | **13.9**±0.2 | **15.8**±0.7 | **15.9**±0.1 | 14.9 |

Table 5: Effectiveness of cross-architecture training models demonstrated after sequentially adding auxiliary NNs.

### 4.5 ABLATION STUDY

We distilled CIFAR-10 by adding different loss function components. From the experimental results in Table 3, it can be seen that $\mathrm{var}\,(\tilde{c}_s)$ has the greatest effect, while $L_{ai}$, $L_{pos}$ have the secondary effect. The benefit of $\mathrm{var}\,(\tilde{c}_s)$ comes from obtaining consistent features recognized by different architectures, $L_{pos}$ merely makes the CAM features more visible, and $L_{ai}$ enable the architecture to benefit from knowledge transferring.

### 4.6 HOW THE NUMBER OF AUXILIARY MODELS INFLUENCES METADD

In this subsection, we investigate the impact of varying the number of auxiliary models on the efficacy of MetaDD. We sequentially add ResNet34, MobileNetV2, GoogleNet, and ViT-B-16 to MetaDD without retrieval. With each addition of an auxiliary NN, we conduct cross-model generalization experiments on ILSVRC-2012.

The experimental results, as shown in Table 5, indicate that as the number of auxiliary models increases, the performance of MetaDD improves on both seen and unseen model architectures. The improvement is particularly pronounced when adding models from the same series. Therefore, for MetaDD, including a diverse set of auxiliary models with significant structural differences enhances generalization.

## 5 CONCLUSION

We introduce MetaDD, a new component specifically designed to enhance the cross-architecture generalizability of DD. MetaDD delivers the dual advantages of minimal additional computational overhead and improved performance. By delving into the factors that limit cross-architecture generalizability, MetaDD uncovers the unique feature recognition mechanisms inherent to different neural network architectures, which often prioritize diverse and heterogeneous features. However, these architectures also adhere to certain shared aesthetic or structural standards. MetaDD enhances cross-architecture generalizability by amplifying the representation of meta features that align with these shared standards. It achieves this by synthesizing meta features through the integration of unified CAM outputs from various neural networks, ensuring these meta features are broadly recognized and effectively utilized across different architectures.

## 6 ETHICS STATEMENT

In this study, we adhere to the ICLR Code of Ethics, ensuring that all aspects of our research meet ethical standards. Our research does not involve human subjects, thus no Institutional Review Board (IRB) approval is required. The datasets utilized are publicly available, and we follow best practices for data release, giving appropriate credit in our citations.

We acknowledge that machine learning models can introduce biases. Therefore, we have carefully examined fairness and potential biases during model design and evaluation. Our experiments include a thorough analysis of model performance across diverse populations and conditions, with discussions included in our results.

In summary, we are committed to conducting our research responsibly, ensuring that all processes comply with research integrity and legal requirements.

## 7 REPRODUCIBILITY

We provide the hyperparameter settings for all dataset configurations in the appendix. And we will release our code shortly.

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

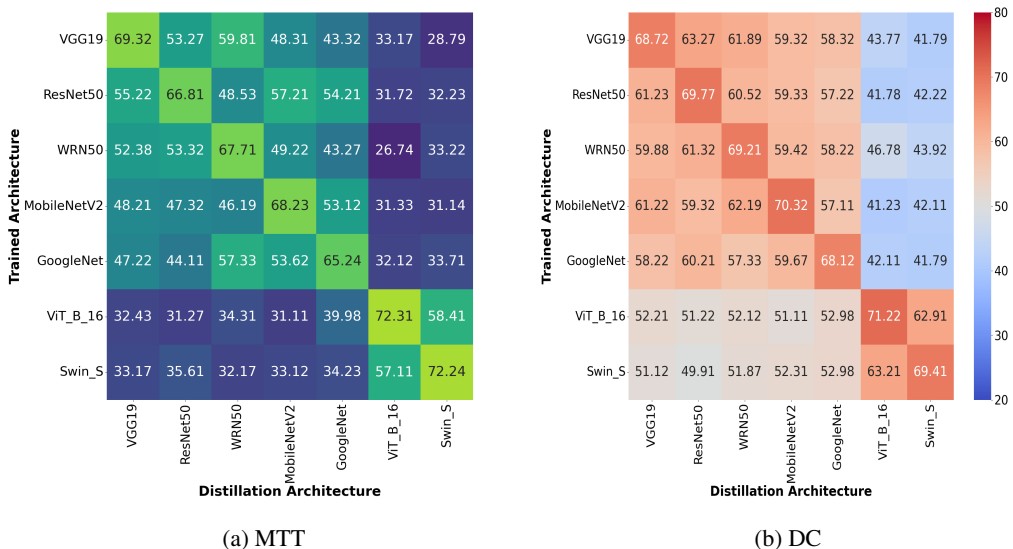

(a) MTT

(b) DC

Figure 6: The top-1 accuracy on the test set.

Muxin Zhou, Zeyuan Yin, Shitong Shao, and Zhiqiang Shen. Self-supervised dataset distillation: A good compression is all you need. *arXiv preprint arXiv:2404.07976*, 2024c.

Lianghui Zhu, Yingyue Li, Jiemin Fang, Yan Liu, Hao Xin, Wenyu Liu, and Xinggang Wang. Weaktr: Exploring plain vision transformer for weakly-supervised semantic segmentation. *arXiv preprint arXiv:2304.01184*, 2023.

# A APPENDIX

## A.1 CROSS-ARCHITECTURE GENERALIZATION GAP PHENOMENON

Using DC and MTT with IPC=10, CIFAR10 was distilled based on VGG19, ResNet50, WRN50, MobileNetV2, GoogleNet, ViT-B-16, and Swin-S. These NNs were then trained. As illustrated in Figure 6, DC Zhao et al. (2020) and MTT Cazenavette et al. (2022) consistently exhibit a notable decline in performance during cross-architecture training on CIFAR10, especially for CNNs and ViT Dosovitskiy et al. (2020). For instance, using a synthetic dataset based on ResNet18 to train MobileNetV2 (a different architecture) yields significantly worse results compared to training ResNet18 (the same architecture). Yet, when using the complete original dataset, MobileNetV2 tends to outperform ResNet18. The cross-architecture transfer gap is even more pronounced between ViT and CNN models than within CNN models alone.

## A.2 HOW THE NUMBER OF AUXILIARY MODELS INFLUENCES METADD

In this subsection, we investigate the impact of varying the number of auxiliary models on the efficacy of MetaDD. We sequentially add ResNet34, MobileNetV2, GoogleNet, and ViT-B-16 to MetaDD without retrieval. With each addition of an auxiliary NN, we conduct cross-model generalization experiments on TinyImageNet and ILSVRC-2012.

The experimental results, as shown in Table 5, indicate that as the number of auxiliary models increases, the performance of MetaDD improves on both seen and unseen model architectures. The improvement is particularly pronounced when adding models from the same series. Therefore, for MetaDD, including a diverse set of auxiliary models with significant structural differences enhances generalization.

| config | value |
|---|---|
| optimizer | Adam |
| base learning rate | 0.01 |
| momentum | 0.9 |
| weight decay | 5e-4 |
| batch size | 200 |
| learning rate schedule | cosine decay |
| training iterator | 1000 |
| epoch per iterater | 100 |
| augmentation | RandomCrop |

(a) MetaDD in DC setting.

| config | value |
|---|---|
| optimizer | Adam |
| base learning rate | 0.01 |
| momentum | 0.9 |
| weight decay | 5e-4 |
| batch size | 200 |
| learning rate schedule | cosine decay |
| training iterator | 500 |
| epoch per iterater | 100 |
| augmentation | RandomCrop |

(b) MetaDD in DM setting.

| config | value |
|---|---|
| optimizer | Adam |
| base learning rate | 0.1 |
| momentum | 0.9 |
| weight decay | 5e-4 |
| batch size | 100 |
| learning rate schedule | cosine decay |
| training epoch | 200 |
| augmentation | RandomCrop |

(c) MetaDD in MTT setting.

| config | value |
|---|---|
| optimizer | Adam |
| base learning rate | 0.25 |
| momentum | 0.9 |
| weight decay | 5e-4 |
| batch size | 200 |
| learning rate schedule | cosine decay |
| training epoch | 200 |
| augmentation | RandomCrop |

(d) MetaDD in Sre2L setting.

Table 6: Hyper-parameter settings.

## A.3 THE METADD HYPERPARAMETER IN DIFFERENT METHODS

As shown in Table 6, we provide hyper-parameter settings for MetaDD in different DD methods.

