# OpenReview forum: "MetaDD: Boosting Dataset Distillation with Neural Network Architecture-Invariant Generalization"
_ICLR.cc/2025/Conference — ICLR 2025 Conference Withdrawn Submission_

### Official Review · Reviewer_CnEy · 2024-10-23

**Soundness:** 2
**Presentation:** 1
**Contribution:** 2
**Rating:** 3
**Confidence:** 4

**Summary:**

This paper primarily focuses on enhancing the generalization performance of domain decomposition (DD) across different neural network architectures. Specifically, the authors decompose features into meta features and heterogeneous features, aiming to maximize the proportion of meta features while minimizing the contribution of heterogeneous features. This process is achieved by adjusting the activation function accordingly. Moreover, to reduce memory consumption, the method involves persistently freezing various layers in the neural network architectures during the DD process. As a result, the approach demonstrates significant improvements in performance.

**Strengths:**

The exploration of cross-architecture generalization, which has been rarely addressed in previous works, opens up a new direction in the field.

Despite introducing multiple auxiliary networks, the method maintains memory efficiency, making it a notable contribution.

MetaDD can also be seamlessly integrated into existing dataset distillation techniques without altering their core mechanisms.

Furthermore, the use of Class Activation Maps (CAMs) to visualize and analyze the features responsible for cross-architecture generalization enhances the interpretability of the approach, adding significant value to the paper.

**Weaknesses:**

The performance improvements are marginal, and in some cases, they can be attributed to variance rather than significant gains. The ablation study shows a similar trend—introducing $L_{ai}$ yields improvements that could be due to errors, particularly in the SRe2L method.

This reliance on pre-trained NNs could limit its flexibility.

The number of experiments conducted on different datasets is somewhat limited. It would be beneficial to validate the model's performance across a broader range of datasets to strengthen the conclusions.

Additionally, the ablation study is not sufficiently comprehensive. It may be helpful to explore different combinations of losses in the ablation study to provide deeper insights into their individual and collective impacts.

**Questions:**

In Fig1, shouldn't the CAM visualizations for cross-arch and same-arch be identical? Since the original dataset hasn't undergone the DD process and is directly tested on a single architecture, there shouldn’t be a distinction between different architectures.

The table of hyperparameters does not clarify the proportions of the individual losses. Are they set to a 1:1:1 ratio? This could potentially diminish the optimization benefits originally intended in the DD process.

There are labeling errors, such as in Table 1(a) under SRe2L seen, where MetaDD and GLaD are listed, but GLaD should be bolded.
Table 2 does not specify which network was used for the experiments, and Table 4 fails to indicate the type of GPU used for running the timing experiments.

A possible issue is that the paper exceeds the 10-page limit.

The layout of various figures and tables is not aesthetically pleasing.

---

### Official Review · Reviewer_EjKe · 2024-10-29

**Soundness:** 2
**Presentation:** 1
**Contribution:** 2
**Rating:** 3
**Confidence:** 4

**Summary:**

MetaDD offers a fresh approach to improving dataset distillation by enhancing generalizability across different architectures. By introducing architecture-invariant loss and concentrating on increasing meta features, it achieves notable advancements over current methods, delivering better performance and reducing memory consumption across multiple neural network architectures.

**Strengths:**

- MetaDD is characterized by low memory consumption and can be seamlessly integrated with existing dataset distillation methods.
- It introduces a certain level of innovation by incorporating architecture-invariant loss functions and feature alignment techniques, addressing the issue of poor generalization performance in dataset distillation across different neural network architectures.
- Class Activation Maps (CAMs) are used to visually demonstrate the feature preferences of different neural network architectures, making it more intuitive.

**Weaknesses:**

- The experimental results are not very optimistic, as the improvement in several groups is negligible.
- Multiple different auxiliary neural networks were added to enhance generalization performance, but the paper does not provide a detailed discussion on how to select the appropriate combination of auxiliary networks.
- When handling large datasets, whether the introduction of multiple auxiliary networks and architecture-invariant loss functions will significantly impact training time remains unclear.

**Questions:**

- Is the bold formatting in the tables somewhat inappropriate? The best results under the same experimental setup should be bolded, rather than just the results of your own method.
- How can the appropriate combination of auxiliary networks be effectively selected in practical applications?
- MetaDD introduces architecture-invariant loss functions and feature alignment techniques to maximize meta features. Has there been a detailed analysis of the weights of these loss terms and their impact on the final generalization performance?

---

### Official Review · Reviewer_rUYz · 2024-11-01

**Soundness:** 2
**Presentation:** 1
**Contribution:** 2
**Rating:** 5
**Confidence:** 4

**Summary:**

This work, titled MetaDD, introduces a distillation plugin module designed to enhance the cross-architecture generalizability of distilled datasets. MetaDD promotes the universality of distilled image features across varying architectural frameworks.

**Strengths:**

The analysis of meta and heterogeneous features provides strong motivation for this study. The approach of enhancing cross-architecture generalization is innovative and promising.

**Weaknesses:**

The improvements brought by MetaDD appear modest.

Numerous typographical errors and incorrect formulas are present, and the authors should refine the manuscript for clarity.
Specific issues include:
- Typographical errors in lines 158 and 159: “improve the vision transformer CAMZhu et al. (2023);…”
- Incorrect use of subindices i, j, u, and v in line 176.
- In Table 1(a), *Sre2L + MetaDD* performs worse than *Sre2L+GLaD* on GoogleNet and ViT-B-16; however, *Sre2L + MetaDD* is incorrectly highlighted in bold.
- In lines 478 to 480, “our method reduces memory usage by $2x$ compared to GLaD”. The author should use $2 \times$ instead of $2x$

**Questions:**

- The authors state that “MetaDD increases meta features and reduces heterogeneous features in distilled data.” Could the authors provide quantitative data on the proportion of area occupied by meta features to substantiate this claim?
- In line 314, “Q and K are the query and key matrices.” However, the reviewer notes that equation (7) does not include Q and K but instead utilizes W_{head}. Could the authors elaborate?
- The column name in Table 5. Does the column “component” means “the number of auxiliary models” and “Quantity” means “method”?
- In Equation (9), should there be a summation over $m$ in the loss function?

---

### Official Review · Reviewer_A4qJ · 2024-11-04

**Soundness:** 3
**Presentation:** 3
**Contribution:** 4
**Rating:** 8
**Confidence:** 4

**Summary:**

This paper introduces the MetaDD framework to improve cross-architecture generalization in dataset distillation (DD) methods. Synthetic datasets generated from specific NN architectures show more heterogeneous features and fewer meta-features, reducing cross-architecture performance. MetaDD addresses this with an architecture-invariant loss, aligning and diversifying GradCAM outputs to optimize feature distribution. Using cross-entropy and Kullback-Leibler divergence terms, this approach enhances diversity and consistency across networks. Experiments show MetaDD outperforms current methods, validating its effectiveness in boosting meta-feature recognition and cross-architecture consistency.

**Strengths:**

- This paper introduces **MetaDD**, a framework designed to significantly improve dataset distillation (DD) by addressing the challenge of cross-architecture generalization, which remains a critical limitation in existing DD methods. By enabling distilled datasets to generalize effectively across various neural network architectures, MetaDD offers a promising solution to one of the primary bottlenecks in DD research.

- **MetaDD’s key innovation** lies in its integration of class activation maps (CAM) into DD algorithms, providing a novel approach to analyzing and understanding different types of features in synthetic datasets. This integration not only enhances the interpretability of distilled data but also strengthens generalizability by aligning feature distributions across architectures, enabling the framework to improve model performance on a variety of architectures.

- **Flexible Integration**: MetaDD is designed as a modular, adaptable component that can be seamlessly incorporated into a range of DD algorithms, enhancing their effectiveness without requiring substantial modifications to the underlying architecture. This flexibility makes MetaDD highly versatile and applicable to diverse DD applications, promoting broader adoption across different neural network models.

- **Technical Rigor**: The framework is supported by robust empirical evidence and thorough analysis, demonstrating its soundness and reliability. Each claim is backed by theoretical underpinnings and practical results, reinforcing MetaDD’s credibility as an effective solution for enhancing DD performance.

- **Performance and Transferability**: Experimental findings show that MetaDD significantly improves the transferability of distilled data, maintaining high performance even on previously unseen architectures. This improved transferability marks a step forward for DD methods, extending their usability and robustness in real-world applications where unseen architectures are common.

- **Clear Structure and Reproducibility**: The paper is well-organized and provides detailed methodology, ensuring that expert readers can easily reproduce the results. This clarity promotes transparency and encourages further exploration by other researchers in the field.

- **Contribution to the Field**: MetaDD advances the state of the art in dataset distillation by effectively increasing the representation of meta-features while reducing heterogeneous features in distilled data, creating a more balanced and efficient feature set for model training. This balance enhances the utility of distilled datasets and lays the groundwork for further improvements in generalization.

- **Balanced Assessment**: The authors present an honest and well-rounded view of MetaDD’s capabilities, highlighting both its strengths and areas for potential improvement. This balanced perspective not only builds trust in the research but also offers valuable insights for future work, positioning MetaDD as a strong foundation for advancing dataset distillation techniques.

**Weaknesses:**

- **Inappropriate Comparison Selection**: The paper compares the Dream algorithm with MetaDD, but since Dream is not designed to address cross-architecture generalization, this comparison lacks a sound basis. More persuasive comparisons could be made with algorithms that share similar objectives.

- **Inadequate Explanation of Loss Function Components**: The individual roles of each component within the loss function are not thoroughly explained, resulting in an unclear presentation of the design rationale. It would be beneficial to detail the contribution of each component to the optimization process, clarifying the method's effectiveness.

- **Unclear Relationship Between Meta-Features and Heterogeneous Features**: The paper does not clearly explain why an increase in meta-features would result in a reduction of heterogeneous features. Additional theoretical or empirical evidence on this relationship would strengthen the method's scientific validity and coherence.

**Questions:**

1. Why is the Dream algorithm used for comparison with MetaDD, despite not being designed for cross-architecture generalization? Would a comparison with more relevant algorithms be more suitable?

2. Could the roles of each component in the loss function be explained in more detail? How does each component contribute to the overall optimization process?

3. Why does an increase in meta-features result in a reduction of heterogeneous features? Could additional theoretical or empirical support clarify this relationship?

---

### Note · Authors · 2024-11-13

I have read and agree with the venue's withdrawal policy on behalf of myself and my co-authors.